# Suppression of Food Allergic Symptoms by Raw Cow’s Milk in Mice is Retained after Skimming but Abolished after Heating the Milk—A Promising Contribution of Alkaline Phosphatase

**DOI:** 10.3390/nu11071499

**Published:** 2019-06-30

**Authors:** Suzanne Abbring, Joseph Thomas Ryan, Mara A.P. Diks, Gert Hols, Johan Garssen, Betty C.A.M. van Esch

**Affiliations:** 1Division of Pharmacology, Utrecht Institute for Pharmaceutical Sciences, Faculty of Science, Utrecht University, 3584 CG Utrecht, The Netherlands; 2Danone Nutricia Research, 3584 CT Utrecht, The Netherlands

**Keywords:** alkaline phosphatase, allergic diseases, food allergy, immune regulation, milk processing, raw cow’s milk

## Abstract

Raw cow’s milk was previously shown to suppress allergic symptoms in a murine model for food allergy. In the present study, we investigated the contribution of fat content and heat-sensitive milk components to this allergy-protective effect. In addition, we determined the potency of alkaline phosphatase (ALP), a heat-sensitive raw milk component, to affect the allergic response. C3H/HeOuJ mice were treated with raw milk, pasteurized milk, skimmed raw milk, pasteurized milk spiked with ALP, or phosphate-buffered saline for eight days prior to sensitization and challenge with ovalbumin (OVA). Effects of these milk types on the allergic response were subsequently assessed. Similar to raw milk, skimmed raw milk suppressed food allergic symptoms, demonstrated by a reduced acute allergic skin response and low levels of OVA-specific IgE and Th2-related cytokines. This protective effect was accompanied by an induction of CD103^+^CD11b^+^ dendritic cells and TGF-β-producing regulatory T cells in the mesenteric lymph nodes. Pasteurized milk was not protective but adding ALP restored the allergy-protective effect. Not the fat content, but the heat-sensitive components are responsible for the allergy-protective effects of raw cow’s milk. Adding ALP to heat-treated milk might be an interesting alternative to raw cow’s milk consumption, as spiking pasteurized milk with ALP restored the protective effects.

## 1. Introduction

Breastfeeding is the gold standard of infant nutrition. It is a complex matrix providing a unique combination of lipids, carbohydrates, proteins, vitamins and minerals. In addition, breast milk contains numerous components with immunomodulatory properties, such as immunoglobulins, lactoferrin, oligosaccharides, long-chain fatty acids, antioxidants and anti-inflammatory cytokines [1]. These bioactive components are potentially responsible for the allergy-protective effects associated with breastfeeding [2,3,4].

In analogy to breast milk, numerous epidemiological studies have shown that the consumption of raw, unprocessed, cow’s milk can also reduce the risk of allergic diseases [5,6,7,8,9]. These epidemiological findings were recently confirmed by causal evidence, showing that raw cow’s milk prevents the development of house dust mite-induced allergic asthma [10] and of OVA-induced food allergy [11] in murine animal models. However, due to the possible contamination with pathogens, raw cow’s milk consumption is discouraged by regulatory authorities [12]. Even though risks from certified raw cow’s milk, produced under strict hygienic and microbiological standards, are considered to be low [13], a zero-risk can never be attained. Cow’s milk used for commercial purposes is therefore processed.

Milk processing, i.e., heat treatment and homogenization, ensures microbial safety and increases shelf life. Unfortunately, it also impacts the asthma- and allergy-protective effect of raw cow’s milk [5,10,14]. Milk processing considerably alters raw cow’s milk with most prominent effects on the fat content and heat-sensitive milk components. For both constituents, associations have been found in relation to the asthma- and allergy-protective effects. For the fat content of the milk, effects were mainly attributed to the levels of *n*-3 polyunsaturated fatty acids [14], whereas for the heat-sensitive milk components, the whey protein fraction was found to be associated with a reduced allergy risk [5]. Confirming that these raw milk constituents are indeed responsible for the observed allergy protection by showing causality is crucial. This knowledge will further support the development of mildly processed milk, or the addition of specific raw milk ingredients to heat-treated milk as an alternative to raw milk consumption.

In the current study, we investigated to which extent the fat content of the milk and the heat-sensitive milk components contribute to the allergy-protective effects of raw cow’s milk by examining skimmed raw milk and pasteurized milk, respectively, in a murine ovalbumin (OVA)-induced food allergy model. In addition, we added alkaline phosphatase (ALP), one of the first bioactive raw milk components losing its activity upon heat treatment, to pasteurized milk to assess whether this restores the allergy-protective effect.

## 2. Materials and Methods

### 2.1. Mice

Three-week-old, specific pathogen-free, female C3H/HeOuJ mice, purchased from Charles River Laboratories (Sulzfeld, Germany) were housed in filter-topped makrolon cages (one cage/group, *n* = 6–8/cage) at the animal facility of the Utrecht University (Utrecht, The Netherlands) on a 12 h light/dark cycle with unlimited access to food (“Rat and Mouse Breeder and Grower Expanded”; Special Diet Services, Witham, UK) and water. Upon arrival, mice were randomly allocated to the control and experimental groups and were habituated to the laboratory conditions for one week prior to the start of the study. Animal procedures were approved by the Ethical Committee for Animal Research of the Utrecht University and conducted according to the European Directive 2010/63/EU on the protection of animals used for scientific purposes (AVD108002015346).

### 2.2. Milk Types

Raw cow’s milk was collected from a dairy farm (Macroom, Ireland). After collection, the raw cow’s milk was divided into three aliquots. Aliquot 1 was stored without any treatment at −20 °C until further use (raw milk). Aliquot 2 was heated for 15 s at 78 °C, cooled to 4 °C and then stored at −20 °C until further use (pasteurized milk). Aliquot 3 was skimmed at 55 °C to remove the milk fat, cooled to 4 °C and stored at −20 °C until further use (skimmed milk; 0.1% fat). All milk types were produced for experimental purposes only (Danone Nutricia Research, Utrecht, The Netherlands). On the days of milk treatment (Experimental Days -9 to -2; Figure 1), milks were thawed at room temperature and part of the pasteurized milk was spiked with bovine intestinal ALP (pasteurized milk + ALP; 3 units/0.5 mL pasteurized milk; 10× higher concentration than present in raw cow’s milk). ALP was kindly provided by Prof. Dr. W. Seinen (Utrecht University, Utrecht, The Netherlands).

### 2.3. Animal Procedures

A schematic representation of the experimental design is shown in Figure 1. On Experimental Days 0, 7, 14, 21 and 28, mice (*n* = 8/group) were orally sensitized to 20 mg of the hen’s egg protein OVA (grade V; Sigma-Aldrich, Zwijndrecht, The Netherlands) dissolved in 0.5 mL phosphate-buffered saline (PBS) containing 10 µg cholera toxin (CT; List Biological Laboratories, Campbell, CA, USA) as an adjuvant. The PBS-sensitized control mice (*n* = 6) received CT alone (10 µg/0.5 mL PBS). Prior to sensitization, mice were orally treated by using a blunt needle with 0.5 mL raw milk, pasteurized milk, skimmed raw milk, pasteurized milk spiked with ALP, or PBS (as a control) for eight consecutive days (Days -9 to -2). On Day 27, one day before the last sensitization, a blood sample was drawn via cheek puncture to measure basophil activation. On Day 33, five days after the last sensitization, all mice were challenged intradermally in both ears with OVA (10 µg/20 µL PBS) to determine the acute allergic skin response. On the same day, mice were challenged orally with 50 mg OVA dissolved in 0.5 mL PBS. Sixteen hours after the oral challenge (Day 34), a blood sample was taken, and mice were killed by cervical dislocation.

### 2.4. Evaluation of the Acute Allergic Skin Response

To assess the magnitude of the acute allergic skin response to OVA, mice were intradermally challenged in the ear pinnae of both ears with 10 µg OVA in 20 µL PBS. Ear thickness was measured in duplicate for each ear prior to and 1 h after the intradermal challenge using a digital micrometer (Mitutoyo, Veenendaal, The Netherlands). By subtracting the mean basal ear thickness from the mean ear thickness measured 1 h after the intradermal challenge, the ear swelling (expressed as Δ µm) was calculated. Isoflurane (Abbott, Breda, The Netherlands) was used for inhalation anesthesia to perform the intradermal challenge as well as the ear measurements. Measurements were performed blinded.

### 2.5. Basophil Activation Test

The basophil activation test was performed as described previously [15], with few alterations. Briefly, whole blood was drawn from each mouse via cheek puncture on experimental Day 27 (one day before the last sensitization). Blood samples from two mice were pooled and incubated with RPMI 1640 medium (Lonza, Verviers, Belgium), anti-mouse IgE (0.125 µg/mL; eBioscience, Breda, The Netherlands) or OVA (20 µg/mL; Sigma-Aldrich) for 90 min at 37 °C. Activation was stopped with PBS containing 5 mM EDTA (Thermo Fisher Scientific, Paisley, Scotland). After washing the cells twice with PBS, red blood cells were lysed and fixed using a whole blood lysing reagent kit (Beckman Coulter, Brea, CA, USA) according to the manufacturer’s instructions. Cells were then washed again, and non-specific binding sites were blocked by incubating cells for 15 min on ice with anti-mouse CD16/CD32 (Mouse BD Fc Block; BD Biosciences, Alphen aan de Rijn, The Netherlands). Cells were subsequently stained for 30 min on ice with CD4-PE and CD45R/B220-PE to gate out T cells and B cells and with IgE-FITC and CD49b-APC to select basophils. CD200R-PerCP-eFluor^®^ 710 was used as a marker for basophil activation. All antibodies were purchased from eBioscience. Flow cytometry was performed using FACS Canto II (BD Biosciences) and the results were analyzed using FlowLogic Software (Inivai Technologies, Mentone, Australia). Cut-off gates for positivity were established using the fluorescence-minus-one technique.

### 2.6. Measurement of OVA-Specific Immunoglobulins in Serum

Blood samples collected prior to sacrifice were centrifuged at 10,000 rpm for 10 min and serum was stored at −20 °C until analysis of OVA-specific immunoglobulins by means of ELISA. OVA-specific IgE levels were quantified as described previously [16], with few modifications. Briefly, high binding Costar 9018 plates (Corning Inc., New York, NY, USA) were coated overnight at 4 °C with 2 µg/mL purified rat anti-mouse IgE (BD Biosciences) in carbonate/bicarbonate buffer (0.05 M, pH 9.6; Sigma-Aldrich). The next day, plates were washed, blocked for 1 h with PBS/1% bovine serum albumin (BSA; Sigma-Aldrich) and incubated for 2 h with serum samples at room temperature. After washing, plates were incubated for 1 h with 1 µg/mL OVA coupled to digoxigenin (DIG). Plates were then washed again, followed by 1 h incubation with 300 mU/mL anti-DIG-POD Fab fragments conjugated to horseradish peroxidase (Sigma-Aldrich). After washing again, the reaction was developed using *o*-phenylenediamine (Sigma-Aldrich) and stopped by 4 M H_2_SO_4_. The absorbance was measured at 490 nm using a Benchmark microplate reader (Bio-Rad, Veenendaal, The Netherlands). OVA-specific IgE levels are expressed in arbitrary units, calculated based on a titration curve of pooled sera serving as an internal standard. For OVA-specific IgG1 and IgA, high binding Costar 9018 plates were coated with 20 µg/mL OVA (Sigma-Aldrich) in carbonate/bicarbonate buffer and incubated overnight at 4 °C. After overnight incubation, plates were washed and blocked for 1 h with PBS/1%BSA. Serum samples were then incubated for 2 h at room temperature and after washing, plates were incubated for 1.5 h with biotinylated rat anti--mouse IgG1 or IgA detection antibody (1 µg/mL; BD Biosciences). Plates were subsequently washed, incubated for 45 min with streptavidin--horseradish peroxidase (0.5 μg/mL; Sanquin), washed again and developed as described above for IgE. OVA-specific IgG1 and IgA levels are expressed as OD values.

### 2.7. Spleen, Mesenteric Lymph Nodes (MLN) and Lamina Propria (LP) Cell Isolation

Spleen and MLN single cell suspensions were obtained by crushing tissues through a 70 µm nylon cell strainer using a syringe. Splenocyte suspensions were incubated with lysis buffer (8.3 g NH_4_Cl, 1 g KHC_3_O and 37.2 mg EDTA dissolved in 1 L demi water, filter sterilized) to remove red blood cells. Cell suspensions were resuspended in RPMI 1640 medium (Lonza), supplemented with 10% heat-inactivated fetal bovine serum (FBS; Bodinco, Alkmaar, The Netherlands), penicillin (100 U/mL)/streptomycin (100 µg/mL; Sigma-Aldrich) and β-mercaptoethanol (20 µM; Thermo Fisher Scientific) prior to ex vivo OVA-specific restimulation assays or in PBS/1% BSA (Sigma-Aldrich) prior to cell stainings for flow cytometric analysis. For the isolation of small intestinal LP cells (*n* = 6/group), the small intestine was removed, cleared from fat and Peyer’s patches, opened longitudinally, washed in PBS, and cut into 0.5 cm pieces. To remove epithelial cells and intraepithelial lymphocytes, these pieces were washed using Hank’s Balanced Salt Solution (HBSS; Thermo Fisher Scientific) containing 15 mM HEPES (Thermo Fisher Scientific), pH 7.2, and incubated 4 × 15 min at 37 °C with HBSS/HEPES buffer supplemented with 5 mM EDTA, 10% FBS and penicillin (100 U/mL)/streptomycin (100 µg/mL), pH 7.2. After washing with RPMI 1640 medium containing 5% FBS and penicillin/streptomycin, tissue samples were digested for 2 × 45 min on a plate shaker at 37 °C with RPMI 1640 medium supplemented with 5% FBS, penicillin/streptomycin and 0.5 mg/mL collagenase type VIII (Sigma-Aldrich). To collect lamina propria cells, samples were vortexed for 10 s after each incubation and passed through a 100 µm nylon cell strainer. LP cell suspensions were subsequently washed with HBSS/HEPES and purified using a Percoll^®^ density gradient (pH 7.2; GE Healthcare, Uppsala, Sweden). Purified LP cell suspensions were washed and resuspended in PBS/1% BSA for flow cytometric analysis.

### 2.8. Flow Cytometric Analysis of Immune Cells

Spleen-, MLN-, and LP-derived single cell suspensions (0.5–1 × 10^6^ cells/well) were incubated for 15 min on ice with anti-mouse CD16/CD32 (Mouse BD Fc Block; BD Biosciences) in PBS/1% BSA/5% FBS buffer to block non-specific binding sites. Subsequently, cells were extracellularly stained with CD4-PerCP-Cy5.5, CD69-APC, CXCR3-PE, CD25-Alexa Fluor^®^ 488, F4/80-APC-eFluor^®^ 780, CD11c-PerCP-Cy5.5, CD103-APC, CD11b-PE, MHCII-FITC, CD45-PE-Cy7, CD19-PerCP-Cy5.5, CD45R/B220-FITC, latency-associated peptide (LAP)-PE-Cy7 (all purchased from eBioscience), T1ST2-FITC (MD Bioproducts, St. Paul, MN, USA) or CD138-APC (BD Biosciences) for 30 min on ice. Viable cells were distinguished using Fixable Viability Dye-eFluor^®^ 780 (eBioscience). Cells only stained for extracellular markers were fixed using IC Fixation Buffer (eBioscience). Cells additionally stained with intracellular markers were fixed and permeabilized using the FoxP3 Transcription Factor Staining Buffer Set (eBioscience) according to the manufacturer’s protocol and then stained with FoxP3-PE-Cy7 or -APC (eBioscience). Stained cells were measured on the FACS Canto II (BD Biosciences) and analyzed with FlowLogic Software (Inivai Technologies). To increase LAP expression on the surface of MLN-derived lymphocytes, cells were polyclonally stimulated with anti-CD3 (10 µg/mL)/CD28 (1 µg/mL; eBioscience) for 48 h at 37 °C, 5% CO_2_ prior to staining, and boosted afterwards with leukocyte activation cocktail (BD Biosciences) for 4 h at 37 °C, 5% CO_2_.

### 2.9. Cytokine Measurements after ex vivo OVA-Specific Stimulation of Splenocytes

Single cell splenocyte suspensions (8 × 10^5^ cells/well) were cultured in U-bottom culture plates (Greiner, Frickenhausen, Germany) with either medium or OVA (50 µg/mL) for four days at 37 °C, 5% CO_2_. Culture supernatant was collected and stored at −20 °C until measurements of IFNγ, IL-13 and IL-10 by means of ELISA, as described elsewhere [17].

### 2.10. Short-Chain Fatty Acid (SCFA) Analysis in Caecum

Caecal content was collected, snap-frozen in liquid nitrogen and stored at −80 °C until further analysis. After thawing, samples were homogenized by vortexing and diluted in cold PBS (1:10). Samples were subsequently centrifuged, the supernatant was collected and concentrations of acetic, propionic, butyric, isobutyric, valeric and isovaleric acid were determined as previously described [18] by means of a Shimadzu GC2010 gas chromatograph (Shimadzu Corporation, Kyoto, Japan), using 2-ethylbutyric acid as internal standard.

### 2.11. Statistical Analysis

Data are presented as mean ± SEM, including individual data points, and differences between pre-selected groups were statistically determined with one-way ANOVA followed by a Bonferroni’s multiple comparisons test. For plasma cells in the MLN, log-transformed data were used to obtain normality for one-way ANOVA. For the same reason, OVA-specific IgG1 and IgA levels were square root-transformed. As OVA-specific IgE levels were not normally distributed, data were presented as individual data points in a box-and-whisker Tukey plot and analyzed using Kruskal–Wallis test followed by a Dunn’s multiple comparisons test for pre-selected groups. All statistical analyses were performed using GraphPad Prism software (version 7.03; GraphPad Software, San Diego, CA, USA) and results were considered statistically significant when *P* < 0.05.

## 3. Results

### 3.1. Suppression of the Allergic Effector Response by Raw Milk is Retained after Skimming but Abolished after Heating the Milk

To determine whether milk processing affects the capacity of raw cow’s milk to induce tolerance to a non-milk, food allergen, mice were orally treated with raw milk, pasteurized milk or skimmed milk before being sensitized and challenged with OVA. As expected, OVA-sensitized allergic mice showed an increased acute allergic skin response upon intradermal challenge compared to PBS-sensitized control mice (Figure 2A). Exposing mice to raw milk before OVA-sensitization significantly reduced the acute allergic skin response compared to PBS-treated allergic mice (Figure 2A). This protective effect was retained after skimming but abolished after pasteurization of the milk (Figure 2A). Since ALP is one of the first bioactive raw milk components losing activity upon heat treatment, we investigated whether spiking pasteurized milk with ALP would restore the allergy-protective effect. Interestingly, addition of ALP to pasteurized milk significantly lowered the acute allergic skin response compared to PBS-treated allergic mice and pasteurized milk-treated mice (Figure 2A). To study the extent of basophil activation, basophil surface expression of CD200R after stimulation of whole blood with OVA was determined. Even though no difference was observed in CD200R expression on basophils of OVA-sensitized allergic mice compared to PBS-sensitized control mice, CD200R expression was significantly reduced on basophils of mice treated with pasteurized milk + ALP compared to mice treated with pasteurized milk alone (Figure 2B), which is in line with the effects observed on the acute allergic skin response (Figure 2A). OVA-specific IgE levels and plasma cells were not significantly affected by exposure to the different milk types, but they did follow a similar pattern as the acute allergic skin response, with low levels in the raw milk, skimmed milk and pasteurized milk + ALP group and higher levels in the pasteurized milk group (Figure 2C,D). Unfortunately, OVA-specific IgE levels were not significantly increased in OVA-sensitized allergic mice compared to PBS-sensitized control mice (Figure 2C). However, OVA-specific IgG1 and IgA levels did (tend to) increase in these mice, demonstrating an immune response to OVA and supporting sensitization (Figure 2E,F). Functionality of IgE antibodies was furthermore confirmed using a murine bone marrow-derived mast cell degranulation assay (data not shown). For OVA-specific IgG1 and IgA, no differences between milk groups were observed (Figure 2E,F).

### 3.2. Low Th2-Related Cytokine Production by Splenocytes from Raw Milk- and Skimmed Milk-Treated Mice after ex vivo Stimulation with OVA

To investigate whether different milk types affect T helper cell phenotype, spleen and MLN cells were isolated and analyzed by flow cytometry. Percentages of Th1 and Th2 cells were not affected in OVA-sensitized allergic mice compared to PBS-sensitized control mice (Figure 3A–D). However, in the spleen of mice treated with pasteurized milk, Th1 cells tended to decrease compared to allergic mice treated with PBS (Figure 3A). Th2 cell frequency in the spleen did not differ between milk groups (Figure 3B). In the MLN, the percentage of Th1 cells did not differ between milk groups, whereas Th2 cell frequency was increased in mice treated with pasteurized milk + ALP compared to allergic mice treated with PBS (Figure 3D). To determine the functional response of splenocytes and MLN cells upon exposure to OVA, cytokine production was determined. Th1-related IFNγ production by splenocytes was not affected by the different milk types (Figure 3E). For the Th2-related cytokine IL-13, low concentrations were observed in the raw milk, skimmed milk and pasteurized milk + ALP group (Figure 3F), which coincided with the effects observed on the acute allergic skin response (Figure 2A). Pasteurized milk treatment tended to increase the IL-13 production compared to raw milk treatment, whereas adding ALP to pasteurized milk tended to restore the low IL-13 levels (Figure 3F). Compared to raw milk, pasteurized milk also increased the production of IL-10 (Figure 3G), which was previously shown to act as a Th2 cytokine in this OVA-induced food allergy model [19]. Ex vivo stimulation of MLN cells with OVA did not induce detectable cytokine production (data not shown).

### 3.3. Raw Milk and Skimmed Milk Induce Tolerance-Associated Cell Types in the MLN

To assess whether the prevention of OVA-induced food allergic symptoms by raw milk, skimmed milk and pasteurized milk + ALP was associated with the induction of tolerance-associated cell types, changes in different dendritic cell (DC) and regulatory T cell (Treg) subsets were determined in the MLN. DC (CD11c^+^MHCII^+^) numbers tended to increase in raw milk-treated mice and increased in skimmed milk-treated mice compared to PBS-treated allergic mice (Figure 4A). More specific assessment of the DC subsets affected, revealed that both milk types mainly increased the tolerogenic CD103^+^CD11b^+^ DC subpopulation (Figure 4B). Although CD103^+^ DCs are known for their capacity to induce FoxP3^+^ Tregs in the MLN [20], no differences between groups were observed in the percentage of CD25^+^FoxP3^+^ Treg cells (Figure 4C). However, interestingly, the Treg subtype secreting TGF-β, also known as Th3 cells, tended to increase in the raw milk group compared to the pasteurized milk group (Figure 4D).

### 3.4. Increased Percentage of Tolerogenic DCs in MLN of Raw Milk- and Skimmed Milk-Treated Mice is Not Associated with Increased Treg Cell Frequency in the LP

Besides promoting the differentiation of naïve T cells into Treg cells, CD103^+^ DCs also induce the expression of gut-homing receptors on the surface of Treg cells [20]. To investigate whether the increased tolerogenic CD103^+^CD11b^+^ DC subpopulation in the MLN of raw milk- and skimmed milk-treated mice was associated with increased Treg cell trafficking to the gut, lamina propria cells were isolated and analyzed by flow cytometry. However, CD25^+^FoxP3^+^ Treg frequency did not differ between milk groups (Figure 5C) and also CD11c^+^MHCII^+^ DCs and the CD103^+^CD11b^+^ subset showed no differences in the LP (Figure 5A,B).

### 3.5. Different Milk Types Did Not Affect SCFA Concentrations

Since modulation of the gut microbiome might be a way in which raw milk, skimmed milk and pasteurized milk + ALP induced tolerance to OVA, metabolic activity of the gut microbiome was assessed by determining SCFA concentrations in the caecum of the mice. Total SCFA concentrations were not significantly different between groups, but skimmed milk- and pasteurized milk + ALP-treated mice showed the highest levels (Figure 6A). Regarding individual SCFA, a similar pattern was observed for butyric acid and acetic acid concentrations, although again differences did not reach significance (Figure 6B,C). For propionic acid, concentrations were comparable in each milk group (Figure 6D).

## 4. Discussion

We previously showed that raw, unprocessed cow’s milk induces tolerance to OVA, an unrelated, non-milk, food allergen, in a murine food allergy model [11]. In the present study, we demonstrated that this protective effect is retained after skimming but abolished after pasteurization of the milk. Similar to raw cow’s milk, skimmed raw milk reduced the acute allergic skin response after intradermal challenge with OVA. This coincided with low levels of OVA-specific IgE and Th2-related cytokines. An increase in CD103^+^CD11b^+^ DCs and TGF-β-producing Treg cells in the MLN, both associated with tolerance induction, might underlie the allergy-protective effects of raw and skimmed raw cow’s milk. In addition, this study provides a first indication that adding ALP to heat-treated milk might be an interesting preventive strategy since spiking pasteurized milk with ALP restored the allergy-protective effects.

Although several epidemiological studies have shown the potency of raw cow’s milk to reduce/prevent allergic diseases [5,6,7,8,9], its consumption is limited due to the potential presence of pathogens. The risks of diseases outbreaks by pathogens such as *Mycobacterium tuberculosis, Listeria*, *Salmonella*, *Campylobacter*, Enterohemorrhagic *Escherichia coli* and Shigatoxigenic *Escherichia coli* are the reason for governmental agencies to prohibit the sale of raw cow’s milk [12]. To prevent these potential risks, milk used for commercial purposes is processed. This means that, upon collection, raw milk undergoes various processing steps such as milk fat standardization, homogenization and heat treatment. These processing steps have profound effects on the milk structure and are shown to be detrimental to the allergy-protective effects [5,10,14].

Milk processing predominantly affects the fat content of the milk and the heat-sensitive milk components [21,22]. Since milk processing also abolishes the allergy-protective effects of raw cow’s milk, this suggests that these constituents contribute to the observed protection. Indeed, both Wijga et al. (2003) and Waser et al. (2006) showed that frequent consumption of products containing milk fat was associated with a reduced asthma risk [8,23]. In addition, Brick et al. (2016) concluded that part of the asthma-protective effect of raw cow’s milk was explained by a higher fat content and, particularly, higher *n*-3 polyunsaturated fatty acids levels compared to shop milk [14]. However, at the same time, there are also studies where the total fat content was not significantly related to asthma [5]. Epidemiological evidence also exists for a potential contribution of heat-sensitive raw milk components. Loss et al. (2011) demonstrated that raw farm milk, but not boiled farm milk, was inversely associated with asthma, hay fever and atopy. The heat-sensitive whey protein fraction of raw milk was implied to underlie these effects [5]. However, since these are all associations, proof of causality is needed to confirm the protective effects of these different raw milk constituents.

In the present study, we therefore investigated the effect of skimmed raw milk and pasteurized milk in a murine OVA-induced food allergy model. Skimmed milk was as allergy-protective as raw milk, suggesting that the fat content of the milk does not contribute to a large extent to the allergy-protective effects of raw cow’s milk. Our results are in contrast with most of the epidemiological findings [8,14,23], emphasizing the importance of demonstrating a cause–effect relationship. On the other hand, the discrepancy could also be caused by the fact that these epidemiological studies mainly focused on asthma, whereas our study focused on food allergy. Different disease pathogenesis might underlie the different outcomes.

In contrast to skimming, pasteurization abolished the allergy-protective effects of raw cow’s milk in the murine food allergy model used. This is in accordance with epidemiological evidence and with our previous results in a murine asthma model, both showing a loss of protection after heat treatment [5,10]. By comparing milk from the same origin, differing in only one processing step we can conclude with certainty that pasteurization is harmful to the allergy-protective capacity of raw cow’s milk. The importance of heat-sensitive milk components, such as proteins, microRNAs and microbes, is thereby emphasized. Particularly, the heat-sensitive whey protein fraction of raw milk is often mentioned as source of the allergy-protective components. The major whey proteins, namely α-lactalbumin, β-lactoglobulin and bovine serum albumin, do not have immunomodulatory functionalities that can directly be linked to the allergy-protective effects of raw cow’s milk, but several less abundant whey proteins such as immunoglobulins, lactoferrin, TGF-β and IL-10 theoretically do [24,25,26].

A first step towards identifying the potential allergy-protective whey proteins was made by Brick et al. (2017), who investigated the effect of processing intensity on immunologically active milk proteins [27]. As expected, a decrease in the number and abundance of detectable native whey proteins was observed with increasing heat load. Interestingly, the subsequent proteomic analysis categorized the milk samples into two clusters; high (boiled, ultra-high temperature and extended shelf life) and no/low heat treatment (raw, skimmed, pasteurized) [27]. Although pasteurized milk clustered together with raw and skimmed milk, indicating similar native protein patterns, it did not confer protection in our study. One could therefore argue that the overall native protein pattern looks similar but that minor differences still have major consequences for the allergy-protective capacity of the milk. One could also argue that even though pasteurization does not lead to denaturation or chemical modifications of whey proteins, it might lead to loss of functionality.

Although the effect of processing intensity on immunologically active whey proteins is very relevant, it does not provide a direct link to allergic diseases. To provide this link, specific whey proteins can be added to heat-treated milk to see whether they could restore the allergy-protective effect. As a first proof-of-concept, we spiked pasteurized milk with ALP and we assessed the effects on OVA-induced food allergic symptoms. ALP is probably best-known for its function in dairy industry as indicator of successful pasteurization. Upon pasteurization, ALP becomes inactivated and loses its activity, making it an ideal indicator of product safety [28]. Since ALP is one of the first bioactive raw milk components losing activity upon heat treatment, it is also a likely allergy-protective candidate. Oral administration of ALP was already shown to be effective in reducing inflammatory diseases [29,30,31,32], but whether it can also affect allergic diseases has, to our knowledge, never been studied.

Surprisingly, ALP was able to fully restore the allergy-protective effect in the food allergy model used. On practically every parameter assessed, ALP added to pasteurized milk showed similar protective effects as raw milk and skimmed raw milk. As this was a first proof-of-concept, 10 times higher ALP concentrations than present in raw cow’s milk were added to pasteurized milk. We can therefore not yet conclude that ALP is the component underlying the allergy-protective effects of raw cow’s milk, but it seems to be a promising candidate to be used as supplement to heat-treated milk.

In addition to the components involved, this study also provides some indication of the underlying mechanisms. The fact that mice orally treated for eight days with raw cow’s milk were protected against OVA-induced allergic symptoms indicates that they developed oral tolerance to OVA. This oral tolerance was induced in the absence of the allergen, demonstrating that raw cow’s milk has the capacity to induce tolerance via generic immunomodulation. The many immunomodulatory components present in raw cow’s milk are likely to create a tolerogenic environment favoring unresponsiveness upon allergen exposure. Raw cow’s milk is hypothesized to promote Treg cell development, to modulate the gut microbiome and to enhance intestinal barrier function [24,25,26]. However, none of these effects have actually been demonstrated after drinking raw milk.

The present study therefore tried to get more insight into some of these proposed mechanisms. Treg cells are identified as key players in inducing and maintaining oral tolerance [33]. However, in our study, the percentage of CD25^+^FoxP3^+^ Treg cells in the MLN was not affected by raw milk treatment. Interestingly, raw milk did increase the Treg subtype secreting TGF-β compared to pasteurized milk. The importance of these Th3 cells is demonstrated in a study showing reduced numbers in the intestine of food allergic children [34].

Induction of FoxP3^+^ Treg cells occurs in the MLN by CD103^+^ DCs under the influence of retinoic acid and TGF-β [35,36]. CD103^+^ DCs originate in the LP and migrate to the MLNs in a CCR7-dependent manner after acquiring antigen [37]. This DC trafficking from the intestinal mucosa to the MLNs is crucial for oral tolerance induction [38,39]. Interestingly, while raw milk exposure did not significantly affect CD25^+^FoxP3^+^ Treg cells in the MLN, it did increase the tolerogenic CD103^+^CD11b^+^ DC subpopulation. Besides promoting the development of FoxP3^+^ Treg cells, these DCs also induce the expression of gut-homing receptors on the cell surface of FoxP3^+^ Treg cells [35], indicating that the Treg cells might have migrated to the gut. However, also in the LP, FoxP3^+^ Treg cell frequency was not increased by raw milk. Examining effects on Treg cell populations directly after raw milk exposure, instead of at the end of the study, might be of importance, since farm milk exposure was previously shown to be associated with increased FoxP3^+^ Treg cell numbers in children [40].

Regarding the potential immune modulation via the gut microbiome, results were not convincing for raw milk. Caecal SCFA concentrations, as indicator of metabolic activity of the gut microbiome, were not altered compared to other milk groups. However, effects on the gut microbiota itself were not assessed and the timing of measuring SCFA levels might also be crucial in this case. Highest SCFA concentrations, particularly butyric acid and acetic acid, were observed after exposure to skimmed raw milk and ALP. Since oral administration of ALP was previously shown to preserve normal gut microbiome homeostasis [41,42,43], it is tempting to speculate that this feature contributes to its allergy-protective effect.

A limitation of the current study is the lack of a significant IgE response in OVA-sensitized allergic mice compared to PBS-sensitized control mice. However, although significance was not reached, most of the animals in the OVA group did show higher OVA-IgE levels than animals in the PBS group. We would like to emphasize that serum IgE levels do not always correlate with the severity of the allergic response and that allergic symptoms are not solely induced by IgE [44,45,46]. The acute allergic skin response is the primary parameter of food allergic symptoms in the validated mouse model used. This response is acknowledged as a true acute allergic response and recognized as a translatable readout [47,48].

In summary, we demonstrated that the suppression of food allergic symptoms by raw cow’s milk is retained after skimming but abolished after pasteurization of the milk. The data presented therefore indicate that not the fat content, but the heat-sensitive milk components are underlying the allergy-protective effects of raw cow’s milk. The protection by raw and skimmed raw cow’s milk was accompanied by an induction of tolerance-associated cell types in the MLN. In addition, we showed that ALP has the capacity to restore the allergy-protective effects abolished by heat treatment. This study thereby provides, for the first time, a direct link between one of the immunologically active whey proteins present in raw cow’s milk and the suppression of allergic symptoms. Although its potency and mechanism of action still need to be determined, ALP is a promising raw milk component to be added to heat-treated milk. Hence, this research represents an attractive preventive strategy for allergic diseases as alternative to raw milk consumption.

## Figures and Tables

**Figure 1 nutrients-11-01499-f001:**
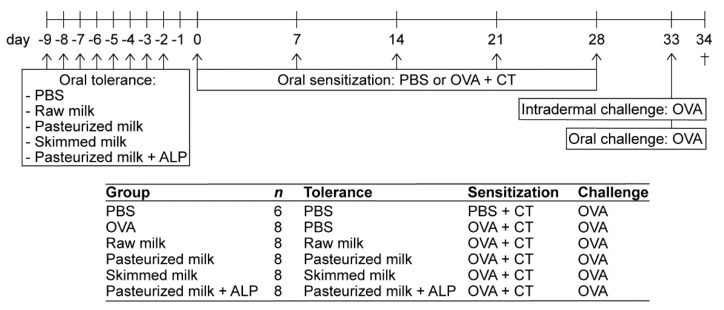
Schematic overview of the experimental setup. Female C3H/HeOuJ mice were randomly allocated to the control and experimental groups: PBS group (PBS-sensitized control mice; *n* = 6), OVA group (OVA-sensitized allergic mice; *n* = 8), raw milk group (raw milk-treated mice; *n* = 8), pasteurized milk group (pasteurized milk-treated mice; *n* = 8), skimmed milk group (skimmed milk-treated mice; *n* = 8) and pasteurized milk + ALP group (pasteurized milk + ALP-treated mice; *n* = 8). Mice were orally treated with 0.5 mL raw milk, pasteurized milk, skimmed milk, pasteurized milk spiked with ALP, or PBS (as a control). Following this oral tolerance induction period, mice were orally sensitized to OVA (20 mg/0.5 mL PBS) with CT as an adjuvant (10 µg/0.5 mL PBS). PBS-sensitized control mice (PBS group) received CT alone. Subsequently, all mice were intradermally and orally challenged with OVA. Mice were killed on day 34 (as indicated by †). PBS, phosphate-buffered saline; ALP, alkaline phosphatase; OVA, ovalbumin; CT, cholera toxin.

**Figure 2 nutrients-11-01499-f002:**
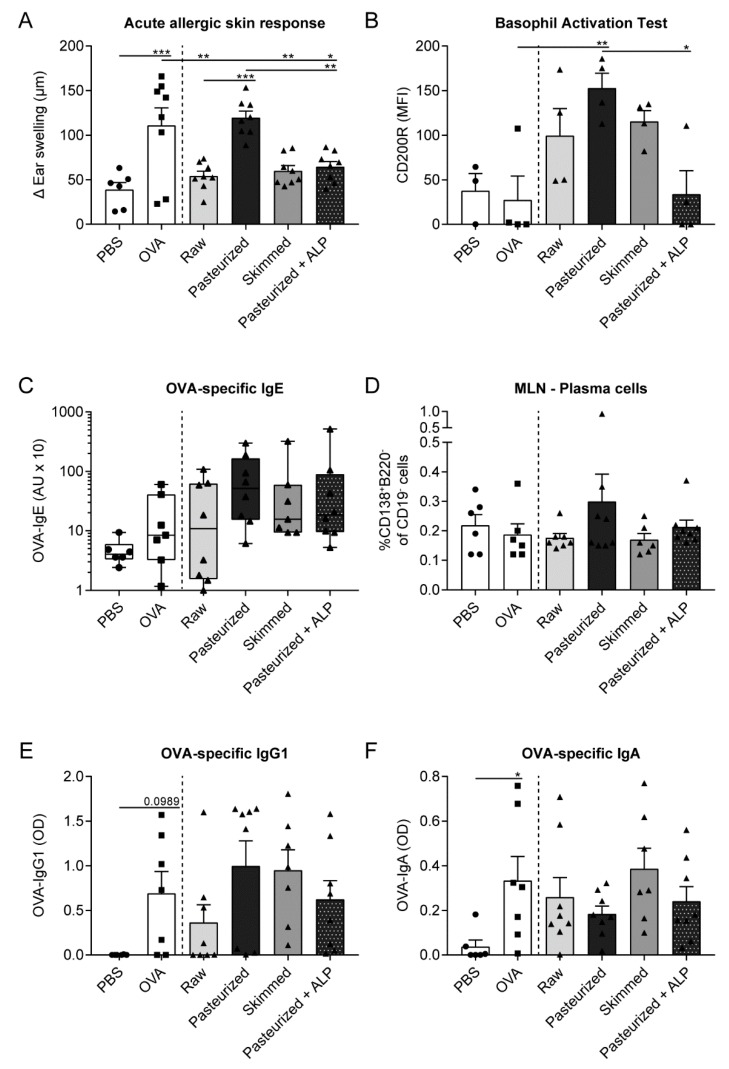
The protective effect of raw milk on the allergic effector response is retained by skimming but abolished by pasteurization of the milk. (**A**) The acute allergic skin response, expressed as Δ ear swelling, measured after intradermal challenge in the ear pinnae of both ears with OVA. (**B**) Basophil activation determined at Day 27 by surface expression of CD200R upon stimulation of whole blood with OVA (after subtracting baseline basophil activation). (**C**) Serum OVA-specific IgE levels measured 16 h after oral challenge. (**D**) Plasma cell (CD138^+^B220^−^ of CD19^−^ cells) frequency assessed in the MLN. (**E**) Serum OVA-specific IgG1 and (**F**) IgA levels measured 16 h after oral challenge. Data are presented as mean ± SEM or as box-and-whisker Tukey plot when data were not normally distributed. In addition, individual data points are displayed, *n* = 6 in PBS group and *n* = 6–8 in all other groups. For the basophil activation test (**B**), blood samples from two mice were pooled, *n* = 3 in the PBS group and *n* = 4 in all other groups. * *P* < 0.05, ** *P* < 0.01, *** *P* < 0.001 as analyzed with one-way ANOVA followed by Bonferroni’s multiple comparisons test for pre-selected groups (**A**,**B**,**D**,**F**) or Kruskal-Wallis test for non-parametric data followed by Dunn’s multiple comparisons test for pre-selected groups (**C**). PBS, phosphate-buffered saline; OVA, ovalbumin; raw, raw cow’s milk; pasteurized, pasteurized cow’s milk; skimmed, skimmed raw cow’s milk; pasteurized + ALP, pasteurized milk spiked with alkaline phosphatase; MFI, median fluorescence intensity; AU, arbitrary units; MLN, mesenteric lymph nodes; OD, optical density.

**Figure 3 nutrients-11-01499-f003:**
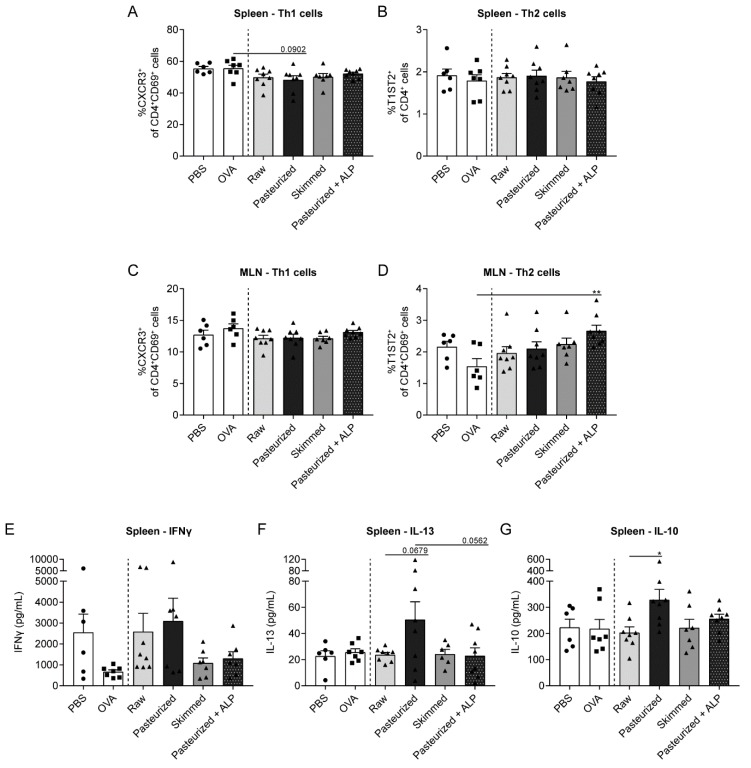
Th2-related cytokine concentrations produced by OVA-stimulated splenocytes were low in raw milk- and skimmed milk-treated mice. (**A**) The percentage of activated Th1 cells (CXCR3^+^ of CD4^+^CD69^+^ cells) and (**B**) Th2 cells (T1ST2^+^ of CD4^+^ cells) in the spleen. (**C**) The percentage of activated Th1 cells (CXCR3^+^ of CD4^+^CD69^+^ cells) and (**D**) activated Th2 cells (T1ST2^+^ of CD4^+^CD69^+^) in the MLN. (**E**) IFNγ, (**F**) IL-13 and (**G**) IL-10 concentrations measured in supernatant of ex vivo stimulated splenocytes with OVA (stimulated for four days, 37 °C, 5% CO_2_). Data are presented as mean ± SEM, including individual data points, *n* = 6 in PBS group and *n* = 6–8 in all other groups. * *P* < 0.05, ** *P* < 0.01 as analyzed with one-way ANOVA followed by Bonferroni’s multiple comparisons test for pre-selected groups. PBS, phosphate-buffered saline; OVA, ovalbumin; raw, raw cow’s milk; pasteurized, pasteurized cow’s milk; skimmed, skimmed raw cow’s milk; pasteurized + ALP, pasteurized milk spiked with alkaline phosphatase; MLN, mesenteric lymph nodes.

**Figure 4 nutrients-11-01499-f004:**
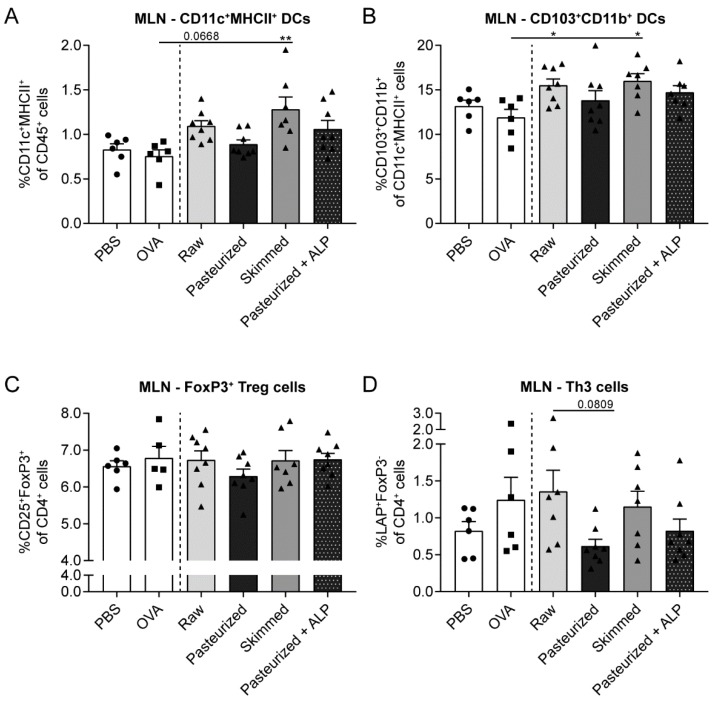
Induction of tolerance-associated cell types in the MLN of raw milk- and skimmed milk-treated mice. (**A**) Percentage of CD11c^+^MHCII^+^ DCs (CD11c^+^MHCII^+^ of CD45^+^ cells), (**B**) CD103^+^CD11b^+^ DCs (CD103^+^CD11b^+^ of CD11c^+^MHCII^+^ cells), (**C**) FoxP3^+^ Treg cells (CD25^+^FoxP3^+^ of CD4^+^ cells) and (**D**) Th3 cells (LAP^+^FoxP3^-^ of CD4^+^ cells) in the MLN. Data are presented as mean ± SEM, including individual data points, *n* = 6 in PBS group and *n* = 5–8 in all other groups. * *P* < 0.05, ** *P* < 0.01 as analyzed with one-way ANOVA followed by Bonferroni’s multiple comparisons test for pre-selected groups. MLN, mesenteric lymph nodes; PBS, phosphate-buffered saline; OVA, ovalbumin; raw, raw cow’s milk; pasteurized, pasteurized cow’s milk; skimmed, skimmed raw cow’s milk; pasteurized + ALP, pasteurized milk spiked with alkaline phosphatase.

**Figure 5 nutrients-11-01499-f005:**
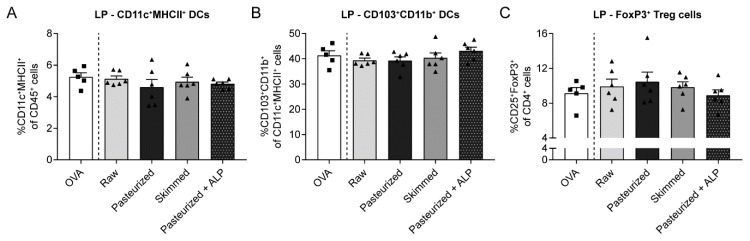
Different milk types did not affect tolerogenic DC and Treg cell frequency in the LP. (**A**) Percentage of CD11c^+^MHCII^+^ DCs (CD11c^+^MHCII^+^ of CD45^+^ cells), (**B**) CD103^+^CD11b^+^ DCs (CD103^+^CD11b^+^ of CD11c^+^MHCII^+^ cells) and (**C**) FoxP3^+^ Treg cells (CD25^+^FoxP3^+^ of CD4^+^ cells) in the LP. Data are presented as mean ± SEM, including individual data points, *n* = 5–6/group. No significant differences were observed. LP, lamina propria; PBS, phosphate-buffered saline; OVA, ovalbumin; raw, raw cow’s milk; pasteurized, pasteurized cow’s milk; skimmed, skimmed raw cow’s milk; pasteurized + ALP, pasteurized milk spiked with alkaline phosphatase.

**Figure 6 nutrients-11-01499-f006:**
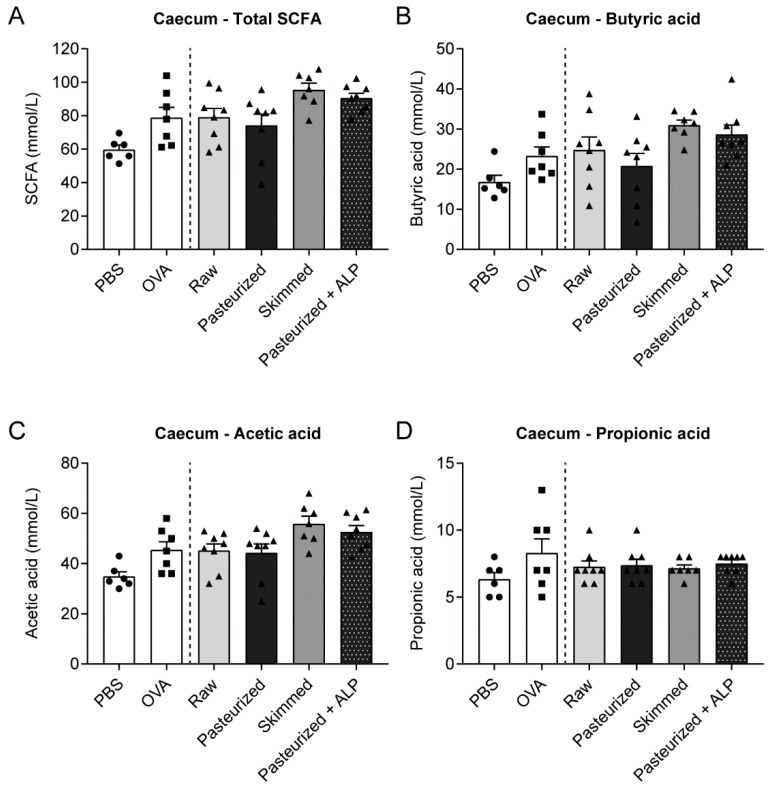
No differences in SCFA concentrations between milk groups. (**A**) Total SCFA concentrations and individual concentrations of (**B**) butyric acid, (**C**) acetic acid and (**D**) propionic acid measured in caecal content. Data are presented as mean ± SEM, including individual data points, *n* = 6 in PBS group and *n* = 7–8 in all other groups. No significant differences were observed. SCFA, short-chain fatty acids; PBS, phosphate-buffered saline; OVA, ovalbumin; raw, raw cow’s milk; pasteurized, pasteurized cow’s milk; skimmed, skimmed raw cow’s milk; pasteurized + ALP, pasteurized milk spiked with alkaline phosphatase.

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
