# Peer review of "Suppression of Food Allergic Symptoms by Raw Cow’s Milk in Mice is Retained after Skimming but Abolished after Heating the Milk—A Promising Contribution of Alkaline Phosphatase"

_nutrients, 2019, doi:10.3390/nu11071499_

Round 1

Reviewer 1 Report

Dear authors,

Your manuscript "Suppression of food allergic symptoms by raw cow’s milk in mice is retained after skimming but abolished after heating the milk – a promising contribution of alkaline phosphatase" deals with an interesting and novel topic. The scientific approach is rigorous. Data are presented clearly. The discussion is wide and complete. I have some minor concerns that you should consider in order to improve some aspects.

GENERAL COMMENT

In general you should try to decrease the number of acronyms in the entire manuscript. Sometimes is better to keep the full name in the entire manuscript, instead of its acronym. This would improve the manuscript fluency.

You choose to submit your manuscript to "Nutrients" Journal, and this is fair. Still, for this reason, you should improve (maybe in the introduction) the description of milk components that might be helpful towards the suppression of food allergic symptoms (e.g. antioxidants, vitamins, proteins, and so on). On this purpose you can refer to "Total antioxidant activity of bovine milk: Phenotypic variation and predictive ability of mid-infrared spectroscopy. International Dairy Journal89, 105-110" and to "Antioxidative peptides derived from milk proteins. International dairy journal16(11), 1306-1314".

SPECIFIC COMMENTS:

Line 20: PBS was not defined.

Line 25: MLN was not defined.

Line 43: It is difficult to verify this citation in order to support your thesis. Consider to change this citation with other in the entire manuscript.

Line 90: was the sensitization "intragastric" (as reported here) or "oral" (as reported in Figure 1)? Choose one term and be consistent in the entire manuscript.

Line 93: why prior to sensitization you decided to treat mice with PBS as control? To me the control group would be no milk (normal diet).

Figure 1: briefly describe the meaning of the table (in the figure caption). If possible, add number of animals, cages and animals/cage. Moreover, the full name of PBS must be provided in figure caption (the full name of PBS is missing also in Figure 2, 3 and following).

Line 116: why did you decide to blend blood samples from two mice?

Line 188: data are presented as "raw" means or as "least squares" means?

Line 189: describe more in details the statistical model, including dependent variables (y) and fixed effect (x) that you considered.

Line 199 to line 201: reformulate (the sentence is too long and hard to read).

Line 240: MLN was previously defined (line 135).

Lines 348-349: You can refer to "Development and validation of an HPLC method for the quantification of tocopherols in different types of commercial cow milk. Journal of dairy science101(8), 6866-6871" to support your statement.

Author Response

The authors would like to thank reviewer 1 for the comments on our manuscript. We address the questions raised by reviewer 1 by answering the specific comments point-by-point.

Reviewer 2 Report

File attached;

Author Response

The authors would like to thank reviewer 2 for the valuable comments on our manuscript. We address the questions raised by reviewer 2 by answering the specific comments point-by-point.

Reviewer 3 Report

Comments

The manuscript „Suppression of food allergic symptoms by raw cow’s milk in mice is retained after skimming but abolished after heating the milk – a promising contribution of alkaline phosphatase”.

In my opinion the manuscript is well written, with aesthetic and informative figures. The most interesting finding is that adding ALP to heat-treated milk might be an interesting alternative to raw cow’s milk consumption, as spiking pasteurized milk with ALP restored the protective effects.

Some comments:

Introduction section

1.      There is also an allergy to breast milk (mother’s diet) (line 33-38)

2.      Raw milk may also be allergic. Should be mentioned in text about the most allergenic components of milk – proteins. What about A1/A2 hypothesis? Genotypes of caseins? The topic of allergy treated a little selectively.

3.      Consumption of raw milk or processed by allergy sufferers has as many supporters as opponents.

4.      The Authors can also inform about baked milk as a way to desensitize.

Materials and methods

Experiment well planned (with informative schematic overwiew), with lots of mice (n=6-8), but I think that you need to add some details in the text:

1.      Why female mice? (line 68)

2.      What kind of food did the mice eat? (line 71)

Author Response

The authors would like to thank reviewer 3 for the comments on our manuscript. We address the questions raised by reviewer 3 by answering the specific comments point-by-point.

Round 2

Reviewer 2 Report

Authors have provided additional data for Total IgE, IL-5 responses and stated that no IL-4 was detected. Provided data show that there was no sensitization to OVA in the mice using the protocol they used. These data, along with the previous data of lack of significant specific IgE, and IL-13 responses to OVA in the mice clearly show that the mice were not sensitized to OVA in the first place.

Authors have agreed that mice did to get sensitized, in their response that 'unfortunately' the positive control did not work.

Immediate hypersensitivity or type I hypersensitivity reaction to food protein like OVA or commonly called as food allergy develops in two stages: 1) sensitization to food protein; and then step 2)  elicitation of acute allergic reaction when challenged with the same food protein (Sampson et al 2018  JACI 141 (1): 11-19). Without sensitization, acute allergic reactions CANNOT occur.

In this case, authors show that their mice showed no evidence of sensitization to OVA as measured by lack of significant specific IgE antibody responses, lack of  total IgE and the underlying cytokine responses such as IL-4, IL-5, IL-13. In other words, the first step required for an an acute allergic reaction did not happen in the mice used in this study.

Therefore, skin swelling observed upon OVA injection to skin in their mice is a non-allergic acute swelling.

Consequently, all interpretation of data with other experiments such as effect of pasteurization, alkaline phosphatase adding etc., must be viewed as NON-ALLERGIC acute skin swelling.

In our experience of doing similar experiments over the past 25 years, unfortunately sometimes some experiments do not simply work and mice do not get sensitized even when standard protocols are used. This thing appears to have happened in their experiments. Therefore, positive controls are essential in all studies. Without sensitization as evidenced by elevated IgE responses, it is not possible to study acute allergic reactions.

Authors must do histopathology of acute skin swelling that they claim has occurred at the site of OVA injection in some mice and show that mast cells have degranulated at the site. Without such direct evidence of mast cell degranulation at the site of injection, and elevated IgE levels in the blood, this study is seriously flawed. Therefore, this study cannot be called as allergy study. It is a non-allergy skin swelling study.

Author Response

The authors would like to thank reviewer 2 for the comments on our manuscript. A response to the comments is provided in the Word document attached.
